

# Optimized Dynamic Mode Decomposition for Reconstruction and Forecasting of Atmospheric Chemistry Data

Meghana Velagar[*], Christoph Keller[**,†], and J. Nathan Kutz[*]

[*]Department of Applied Mathematics, University of Washington, Seattle, WA 98195, USA
[**]NASA Global Modelling and Assimilation Office, Goddard Space Flight Center, Greenbelt, MD, USA
[†]Morgan State University, Baltimore, MD, USA

**Correspondence:** J. Nathan Kutz (EMAIL: kutz@uw.edu)

**Abstract.** We introduce the optimized dynamic mode decomposition algorithm for constructing an adaptive and computationally efficient reduced order model and forecasting tool for global atmospheric chemistry dynamics. By exploiting a low-dimensional set of global spatio-temporal modes, interpretable characterizations of the underlying spatial and temporal scales can be computed. Forecasting is also achieved with a linear model that uses a linear superposition of the dominant spatio-temporal features. The DMD method is demonstrated on three months of global chemistry dynamics data, showing its significant performance in computational speed and interpretability. We show that the presented decomposition method successfully extracts known major features of atmospheric chemistry, such as summertime surface pollution and biomass burning activities. Moreover, the DMD algorithm allows for rapid reconstruction of the underlying linear model, which can then easily accommodate non-stationary data and changes in the dynamics.

## 1 Introduction

The monitoring and forecasting of global atmospheric chemistry is critical for understanding the effects of air quality, chemistry-climate interactions, and global biogeochemical cycling (Jacob, 1999). The dynamics of atmospheric chemistry is characterized by complex interactions among hundreds of chemical species, which can produce kinetics across temporal scales spanning many orders of magnitude, from microseconds to years. Accurate monitoring and prediction requires full knowledge of the chemical state of the atmosphere at all locations and times, resulting in a 5-dimensional data set for longitude, latitude, elevation, species and time that can become massive as the resolution of each dimension is increased. Dimensionality reduction is a critically enabling aspect of machine learning and data science (Brunton and Kutz, 2019) that can be leveraged to approximate the monitoring and forecasting capabilities of global chemistry with more readily tractable computational algorithms (Velegar et al., 2019). *Dynamic mode decomposition* (DMD) is a data-driven regression architecture for adaptively learning linear dynamics models over snapshots of temporal data, specifically in a low-dimensional subspace. DMD has been broadly used in the scientific community due to its ease of use, interpretability and adaptive nature (Kutz et al., 2016a). When applied to the spatio-temporal dynamics of atmospheric chemistry, we demonstrate that the method provides an effective and computational efficient *reduced order modeling* strategy that can be used for characterization, monitoring and forecasting of global chemical





concentrations with either computational or sensor data. Moreover, we show that the optimized DMD algorithm (Askham and

Kutz, 2018) and bagging optimized DMD (BOP-DMD) (Sashidhar and Kutz, 2022) versions of the DMD algorithm are critical
for characterizing the complexities of the chemical interaction dynamics and their uncertainties.

The characterization of multiscale phenomenon, such as that embodied by global atmospheric chemistry, remains challeng-
ing due to the need to resolve spatial and temporal scales that are separated by many orders of magnitude. Computational
methods, which are typically based upon the underlying partial differential equations that model the governing dynamics, eas-

ily become intractable due to the need to resolve the finest space scales and the fastest time scales. Thus, numerical stiffness is
automatically imposed upon a numerical scheme in such a spatio-temporal system. Building models from sensor data directly
is no different: sensors must be placed densely in space in order to resolve spatial features. This also places significant limits
on practicality, as sensors are not only prohibitively expensive, but also require completely impractical global coverage. Com-
putations and sensors, however, are typically used in combination and provide the critical data infrastructure for modeling the

multiscale physics of atmospheric chemistry. So despite the limitations and cost, many advances have been made in our ability
to characterize, predict and monitor global chemistry.

Reduced order models (ROMs) provide an attractive alternative to large scale computing. ROMs provide a mathemati-
cal architecture for reducing the computational complexity of mathematical models in numerical simulations (Benner et al.,
2015; Antoulas, 2005; Quarteroni et al., 2015; Hesthaven et al., 2016). Fundamental to rendering simulations computationally

tractable is the construction of a low-dimensional subspace on which the dynamics can be approximately embedded. Unfortu-
nately, projective-based ROM construction often produces a low-rank model for the dynamics that can be unstable (Carlberg
et al., 2017.), i.e. the models produced generate solutions that rapidly go to infinity in time. Machine learning techniques offer a
diversity of alternative methods for computing the time-dynamics in the low-rank subspace, with a diversity of neural networks
showing how to advance solutions, or learn the flow map from time $t$ to $t + \Delta t$ (Qin et al., 2019; Liu et al., 2020). Indeed, deep

learning algorithms provide a flexible framework for constructing a mapping between successive time steps. The typical ROM
architecture constrains the dynamics to a subspace spanned by POD (proper orthogonal decomposition), thus in the new POD
coordinate system, time evolution can be used to construct a time-stepping model using neural networks. Recently, (Parish and
Carlberg, 2020) and (Regazzoni et al., 2021) developed a suite of neural network based methods for learning time-stepping
models for tropospheric bromine chemistry and cardiovascular dynamics, respectively. Moreover, (Parish and Carlberg, 2020)

provide extensive comparisons between different neural network architectures along with traditional techniques for time-series
modeling.

Projective ROMs are often unstable and ill-suited for massive multiscale systems, while deep learning models require signif-
icant time and data for training and also assume stationarity of the data in order for the results to be valid for withheld test sets.
Both of these limitations make their use in global atmospheric modeling problematic. However, a computationally efficient

and adaptive ROM approach is embodied by DMD. DMD was introduced as an algorithm by (Schmid, 2010) and has rapidly
become a commonly used data-driven analysis tool. It is the leading approximation method for the Koopman (linear) operator
from data (Rowley et al., 2009). DMD by construction provides a method for identifying spatio-temporal coherent structures in
high-dimensional time-series data. DMD analysis offers a dynamic version of standard dimensionality reduction methods such





as the *proper orthogonal decomposition* (POD), which highlighted low-rank features in spatio-temporal data (Kutz, 2013). However, DMD not only provides a low-rank subspace, but each mode is associated with linear (exponential) behavior in time, often given by oscillations at a fixed frequency with growth or decay. Thus, DMD is a regression to solutions of the form

$$\mathbf{x}(t) = \sum_{j=1}^{r} \boldsymbol{\phi}_j e^{\omega_j t} b_j = \boldsymbol{\Phi} \exp(\boldsymbol{\Omega}t)\mathbf{b}, \tag{1}$$

where $\mathbf{x}(t)$ is an $r$-rank approximation to a collection of state space measurements $\mathbf{x}_k = \mathbf{x}(t_k)$ $(k = 1, 2, \cdots, n)$. The algorithm regresses to values of the DMD eigenvalues $\omega_j$, DMD modes $\boldsymbol{\phi}_j$ and their loadings $b_j$. The $\omega_j$ determines the temporal behavior of the system associated with a modal structure $\boldsymbol{\phi}_j$. Such a regression can also be learned from time-series data (Lange et al., 2020). DMD may be thought of as a combination of SVD/POD in space with the Fourier transform in time, combining the strengths of each approach (Chen et al., 2012; Kutz et al., 2016a). DMD is modular due to its simple formulation in terms of linear algebra, resulting in innovations related to control (Proctor et al., 2016; Deem et al., 2020), compression (Erichson et al., 2016; Brunton et al., 2015), reduced-order modeling (Alla and Kutz, 2017), and multi-resolution analysis (Kutz et al., 2016b; Liu et al., 2023), among others.

## 2 Atmospheric Chemistry Data Sets, Data Pre-processing, and Methods

### 2.1 Atmospheric chemistry model

Understanding the composition of the atmosphere is critical for a wide range of applications, including air quality, stratospheric ozone loss, and environmental degradation (Jacob, 1999). Chemical transport models (CTM) are used to simulate the evolution of atmospheric constituents in space and time (Brasseur and Jacob, 2017). A CTM solves the system of coupled continuity equations for an ensemble of $m$ species with number density vector $\mathbf{n} = (n_1, \ldots, n_m)^T$ via operator splitting of transport and local processes:

$$\frac{\partial n_i}{\partial t} = -\nabla \cdot (n_i \mathbf{U}) + (P_i - L_i)(\mathbf{n}) + E_i - D_i \qquad i \in [1, m] \tag{2}$$

with $\mathbf{U}$ being the wind vector, $(P_i - L_i)(\mathbf{n})$ the (local) chemical production and loss terms, $E_i$ the emission rate, and $D_i$ the deposition rate of species $i$. The transport operator,

$$\frac{\partial n_i}{\partial t} = -\nabla \cdot (n_i \mathbf{U}) \qquad i \in [1, m] \tag{3}$$

involves spatial coupling across the model domain but no coupling between chemical species, while the chemical operator,

$$\frac{dn_i}{dt} = (P_i - L_i)(\mathbf{n}) + E_i - D_i \qquad i \in [1, m] \tag{4}$$

includes no spatial coupling but the species are chemically linked through a system of ordinary differential equations (ODEs). Chemistry models repeatedly solve equations (3) and (4), which requires full knowledge of the chemical state of the atmosphere at all locations and times. The resulting 4-dimensional data sets (longitude,latitude,levels,species) can become massive,



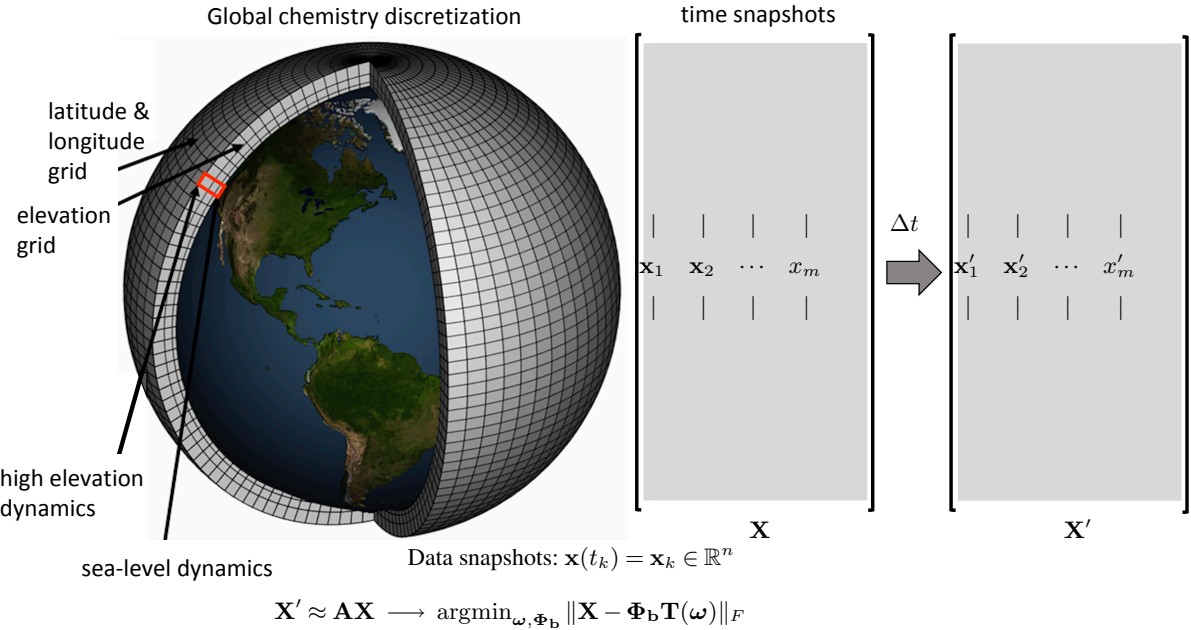

$$\mathbf{X}' \approx \mathbf{A}\mathbf{X} \longrightarrow \mathrm{argmin}_{\boldsymbol{\omega}, \boldsymbol{\Phi_b}} \|\mathbf{X} - \boldsymbol{\Phi_b}\mathbf{T}(\boldsymbol{\omega})\|_F$$

**Figure 1.** *The spatial grid for atmospheric chemistry data sets on the left panel. The data $\mathbf{x}(t_k)$ is collected into snapshot matrices $\mathbf{X}$ which are used to regress to the best exponential (linear) solution $\mathrm{argmin}_{\boldsymbol{\omega}, \boldsymbol{\Phi_b}} \|\mathbf{X} - \boldsymbol{\Phi_b}\mathbf{T}(\boldsymbol{\omega})\|_F$, where $\boldsymbol{\Phi_b}$ are the weighted DMD modes and $\mathbf{T}$ is a matrix of exponentials for fitting the data (6).*

which makes it unpractical to output them at high temporal frequency and refined spatial resolution. As a consequence, model output is generally restricted to a few selected species of interest (e.g. ozone), while the full model state is only output very infrequently, e.g. to archive the information for future model restarts. We show here that the chemical state of a CTM such as GEOS-Chem has distinct low-ranked features and exploiting these properties using modern diagnostic tools such as variable reduction or sub-sampling makes it possible to represent the majority of information in a computationally more efficient manner. While we focus here on identifying low-ranked features across the spatio-temporal dimension (i.e., for each species separately) the presented methods could similarly (and independently) be applied across the species domain.





### 2.1.1 Global Atmospheric Chemistry Simulations

The reference simulation of atmospheric composition was generated using the GEOS-Chem model, as described in (Velegar et al., 2019). GEOS-Chem (https://geoschem.github.io) is an open-source global model of atmospheric chemistry used for a wide range of applications. The model can be run in offline mode as a chemical transport model (CTM) (Bey, 2001; Eastham et al., 2018) or as an online component within the NASA Goddard Earth System Model (GEOS) (Long et al., 2015; Hu et al., 2018). The dataset used here was produced using the offline version of GEOS-Chem (v11-01), driven by archives of

assimilated meteorological data from the GEOS Forward Processing (GEOS-FP) data stream of the NASA Global Modeling and Assimilation Office (GMAO). Model chemistry includes detailed HOx-NOx-VOC-ozone-BrOx tropospheric chemistry as originally described by (Bey, 2001), with addition of BrOx chemistry by (Parrella et al., 2012) and updates to isoprene oxidation as described by (Mao et al., 2013). Stratospheric chemistry is simulated using a linearized mechanism as described by (Murray et al., 2012).

The model output covers one year (July 2013 - June 2014) at $4° \times 5°$ horizontal resolution, providing a comprehensive set of atmospheric chemistry model diagnostics. For every chemistry time step of 20 minutes, the concentrations of all 143 chemical constituents were archived immediately before and after chemistry in units of molecules/cm$^3$. The difference between these concentration pairs are the species tendencies due to chemistry (expressed in units of molecules/cm$^3$/s). Since the solution of chemical kinetics is sensitive to the environment, we further output key environmental variables such as temperature, pressure,

water vapor, and photolysis rates. The latter are computed online by GEOS-Chem using the Fast-JX code of (Bian and Prather, 2002) as implemented in GEOS-Chem by (Mao et al., 2010) and (Eastham et al., 2014). At every time step, the data set thus consists of nfeatures $= 143 + 91 + 3 + 143 = 380$ data points at every grid location. We restrict our analysis to the lowest 30 model levels to avoid influence from the stratosphere. The resulting data set has dimensions nlon × nlat × nlev × ntimes × nfeatures $= 72 \times 46 \times 30 \times 26280 \times 380 = 9.9 \times 10^{11}$.

## 2.2 Data Pre-Processing

Many dimensionality reduction techniques rely on an underlying singular value decomposition of the data that extracts correlated patterns in the data. A fundamental weakness of such SVD-based approaches is the inability to efficiently handle invariances in the data. Specifially, translational and/or rotational invariances of low-rank features in the data are not well captured (Kutz, 2013; Kutz et al., 2016a; Brunton and Kutz, 2019; Velegar et al., 2019). One of the key environmental variables

driving the chemistry is photolysis rate, the absolute concentrations of many chemicals of interest accordingly 'turn on' and are non zero during day time, and 'turn off' or go to zero during the night. The time series of absolute chemical concentrations exhibit a translating wave traversing the globe from east to west with constant velocity. The time series for the chemical species $O_3$ (Ozone) is plotted with respect to UTC time for one latitude $= 30°$/elevation $= 1$ and three different longitudes $= [-100°, 0°, 100°]$ on bottom left in Fig. 2, highlighting the translational invariance in the absolute concentration data. Any

SVD-based approach will be unable to capture this translational invariance and correlate across snapshots in time, producing an artificially high dimensionality, i.e., higher number of modes would be needed to characterize the dynamics due to transla-





tion (Kutz, 2013; Brunton and Kutz, 2019). To overcome this issue the time series for each grid point are shifted to align with the GMT time, as shown on bottom middle in Fig. 2. With the local times for each grid point aligned SVD-based dimensionality reduction techniques can now identify and isolate coherent low-dimensional features in the data. Similarly, the current

season dictates length of days and nights. Latitudes where the days are very short, i.e., the 'turn-on' times are very short, the chemistry exhibits "spiky" patterns. SVD-based approaches would again need an artificially high number of modes to capture the low-rank features in the data. To work around this issue the day time chemistry can be isolated and analysis performed on the isolated day times, especially if there is total 'turn-off' of dynamics during night times. On the bottom right the day time chemistry is isolated showing only the non-zero data during daytime.


### 2.3 Optimized Dynamic Mode Decomposition (DMD)

The DMD algorithm schematic is shown in the right panel of Fig. 1. The DMD algorithm seeks the leading spectral decomposition of the best fit linear operator $\mathbf{A}$ (Brunton and Kutz, 2019) that approximately advances the snapshot measurements of the state of a system $\mathbf{x} \in \mathbb{R}^n$ forward in time by stepsize $\Delta t$:

$$\mathbf{X}' \approx \mathbf{A}\mathbf{X} \tag{5}$$

which leads to the mathematical definition of operator $\mathbf{A}$ as the best fit one-step operator (Tu et al., 2014).

However, the DMD formulated by this regression is rarely used for forecasting and/or reconstruction of time-series data except in cases with noise-free or nearly noise-free data. This is because the exact DMD (5) is extremely sensitive to noise in

the data, causing a bias in the computed DMD modes and eigenvalues (Bagheri, 2014; Dawson et al., 2016; Hemati et al., 2017). The *optimized DMD* algorithm of Askham and Kutz (Askham and Kutz, 2018), which uses a variable projection method (Golub and Pereyra, 2003) for nonlinear least squares to compute the DMD for unevenly timed samples, provides the best and most optimal performance of any algorithm currently available. Indeed, this optimal performance is mathematically guaranteed by the exponential fitting procedure of Askham and Kutz (Askham and Kutz, 2018). The exponential fitting is given

by

$$\mathrm{argmin}_{\omega_k, \boldsymbol{\phi}_k, b_k} \|\mathbf{X} - \sum_{k=1}^{r} b_k \boldsymbol{\phi}_k \exp(\omega_k \mathbf{t})\|_2^2 \tag{6}$$

where a rank $r$ approximation is estimated. As noted, optimized DMD iterates to a solution of this non-convex problem by using variable projection (Golub and Pereyra, 2003). This has been shown to provide a superior decomposition due to its ability to optimally suppress noise bias and handle snapshots collected at arbitrary times. Fig. 3 shows a comparison of surface

nitrogen oxide (NO) as produced by GEOS-Chem (top panel), reconstructed using classical or exact DMD (middle panel), and using optDMD (bottom panel). The classical DMD reconstruction dies out within a few days, failing in the task of even



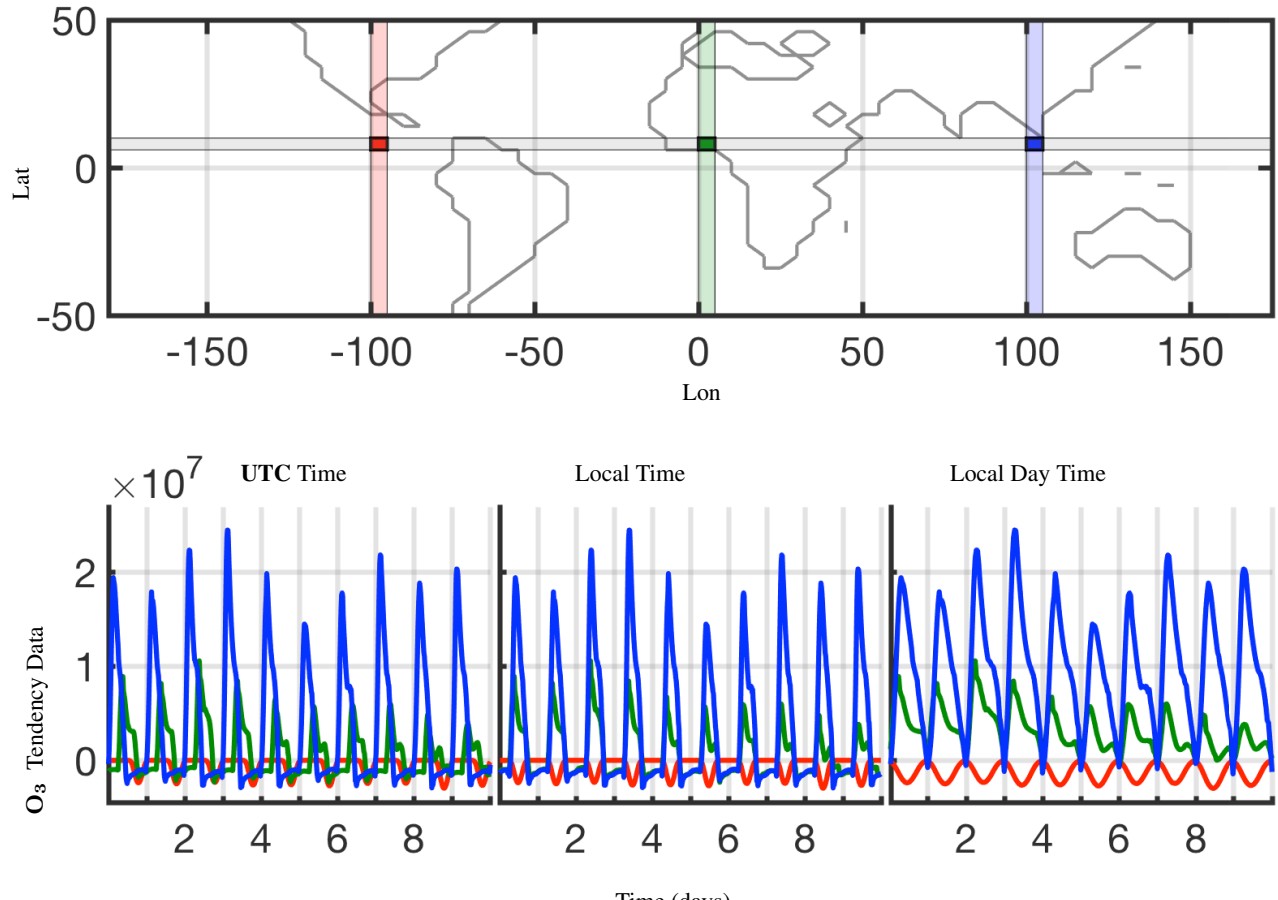

**Figure 2.** *Shifting the data for each cell in time to align the local time zones across a latitude to the prime meridian*$(\mathrm{Lon} = 0°)$ *local time, shown here for* $\mathbf{O_3}$ *tendency data for* $\mathrm{Lat} = 30°$. *The bottom left panel is the raw data for the 3 highlighted cells, the bottom center panel is this data shifted in time, and the bottom right panel shows isolated day time values only.*

reconstructing the time-series data, let alone forecasting, as it was originally regressed to. In contrast, the optDMD is able to capture, sustain and faithfully reconstruct the original time series.

We can also introduce constraints to the optDMD algorithm, including constraining all the DMD eigenvalues in (6) to (i)

The imaginary axis:

$$\text{subject to } \Re(\omega_k) = 0 \qquad\qquad (7)$$



For $\mathbf{NO_{CONC}}$ data



**Figure 3.** *Comparing* 30 *day reconstruction results for Classical and Optimized DMD at the surface of* **NO** *preprocessed data at* Lat $=$ $30°$. *The results are for absolute concentration or* **CONC** *data; the top panel shows the preprocessed data, the middle panel shows the reconstruction from the Classical* DMD*, and the bottom panel shows the reconstruction from Optimized* DMD. *The Classical* DMD *is unable to capture the dynamics for the absolute concentration data and it decays down to zero. The Optimized* DMD *reconstructs the data and resolves the dynamics accurately.*

(ii) The closed left-half plane:

$$\text{subject to } \Re(\omega_k) \leq 0 \tag{8}$$



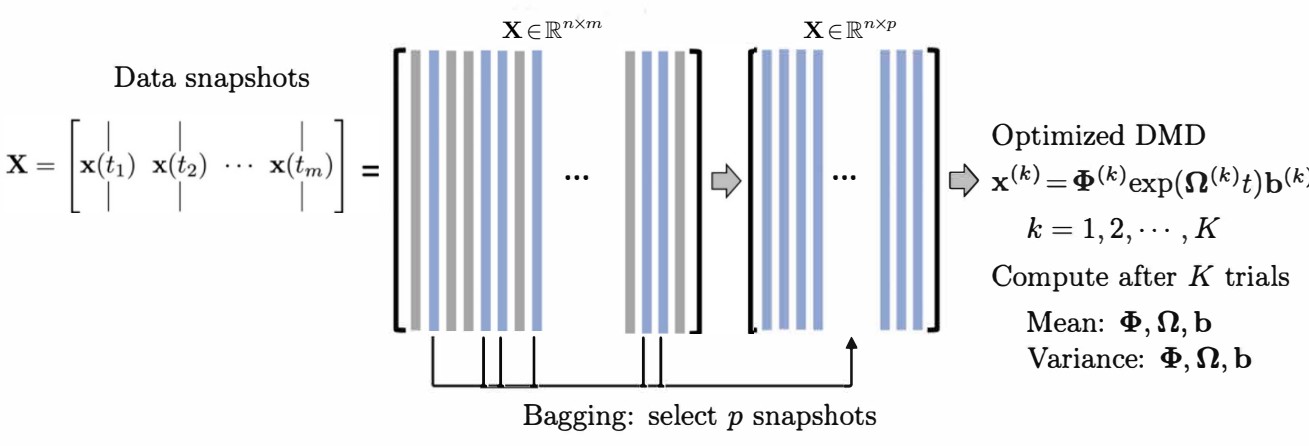

**Figure 4.** *Summary of the BOP-DMD architecture reproduced with permission from (Sashidhar and Kutz, 2022). The data snapshots $\mathbf{x}(t_k)$ are collected over $m$ snapshots into the matrix $\mathbf{X}$. Columns of $\mathbf{X}$ are randomly sub-selected into the matrix $\mathbf{X}^{(k)}$ to build an optimized DMD model. Each DMD model $\mathbf{x}^{(k)} = \mathbf{\Phi}^{(k)} \exp(\mathbf{\Omega}^{(k)}t)\mathbf{b}^{(k)}$ is used to compute the statistics (mean and variance) of the DMD parametrizations $\mathbf{\Phi}$, $\mathbf{\Omega}$, $\mathbf{b}$ which are used in building a the BOP-DMD ensemble solution with Uncertainty Quantification (UQ).*

As discussed below, these constraints further stabilize and make robust reproduction and forecast of the time series data. The

disadvantage of optimized DMD is that one must solve a nonlinear optimization problem through variable projection (Golub and Pereyra, 2003), often which can at times fail to converge.

### 2.4   Bagging OPtimized Dynamic Mode Decomposition (BOP-DMD)

BOP-DMD (Sashidhar and Kutz, 2022) leverages Breiman's statistical bagging sampling strategy (Leo Breiman, 1984) in

partnership with the optimized DMD algorithm. The BOP-DMD architecture is presented in Fig. 4. Bagging is designed to produce an ensemble of models, thereby reducing model variance and suppressing over-fitting by design. Not only does ensembling improve DMD, it also is effective in deep neural network regressions (Allen-Zhu and Li, 2020). Further innovations include stabilizing the variable projection technique used by optDMD so that it converges consistently to an optimal solution (Sashidhar and Kutz, 2022). Its ability to converge is often dependent upon a suitable initial guess for the DMD

eigenvalues and eigenvectors.





The BOP-DMD algorithm accounts for the initialization process and further provides the optimal solutions to linear models by using optDMD as the regression architecture. Algorithm **??** shows the algorithmic structure of BOP-DMD, highlighting the bagging, initialization and ensembling of the DMD models to produce an ensemble, probabilistic DMD model. The initialization of DMD is accomplished by first constructing an optDMD model approximation, whose eigenvalues and eigenvectors

$\boldsymbol{\Phi}_0$ can be used to seed the BOP-DMD. $p$ snapshots are randomly selected from the full data matrix $\mathbf{X} \in \mathbb{R}^{n \times m}$, to form a subset data matrix $\mathbf{X} \in \mathbb{R}^{n \times p}$. optDMD produces the model for this subset data, and we save the resulting model parameters. The process is repeated for $K$ trials producing an ensemble of optDMD models. The mean $\{\langle \boldsymbol{\Phi} \rangle, \langle \boldsymbol{\Omega} \rangle, \langle \mathbf{b} \rangle\}$ and variance $\{\langle \boldsymbol{\Phi}^2 \rangle, \langle \boldsymbol{\Omega}^2 \rangle, \langle \mathbf{b}^2 \rangle\}$ of the model parameters $\boldsymbol{\Phi}$, $\boldsymbol{\Omega}$, $\mathbf{b}$ can now be computed. Hence, in addition to producing the DMD model itself, the output of algorithm **??** generates both spatial and temporal uncertainty quantification metrics or UQ metrics. In this

work we primarily focus on the temporal UQ metrics for forecasting.

**Algorithm 1: BOP-DMD**

**Input:** Input $(\mathbf{X}, p, K)$

   **Procedure:** BOPDMD $(\mathbf{X}, p, K)$

Compute $\boldsymbol{\Phi}_0, \boldsymbol{\Omega}_0, \mathbf{b}_0$

     For $k \in \{1, 2, \cdots, K\}$

          Choose $p$ of $m$ snapshots $(p < m)$

          optDMD $\boldsymbol{\Phi}_k, \boldsymbol{\Omega}_k, \mathbf{b}_k$ and Initialize with $\boldsymbol{\Omega}_0$

          Update $\boldsymbol{\Phi}, \boldsymbol{\Omega}, \mathbf{b}$ by adding $\boldsymbol{\Phi}_k, \boldsymbol{\Omega}_k, \mathbf{b}_k$ to $\boldsymbol{\Phi}, \boldsymbol{\Omega}, \mathbf{b}$

Compute mean $\boldsymbol{\mu} = \{\langle \boldsymbol{\Phi} \rangle, \langle \boldsymbol{\Omega} \rangle, \langle \mathbf{b} \rangle\}$

     Compute variance $\boldsymbol{\sigma} = \{\langle \boldsymbol{\Phi}^2 \rangle, \langle \boldsymbol{\Omega}^2 \rangle, \langle \mathbf{b}^2 \rangle\}$

     **return:** $\boldsymbol{\mu}, \boldsymbol{\sigma}$ which are optDMD parameters.

## 3   Results

The analysis is performed for preprocessed or time-shifted raw data for 60 days, from July, $2^{\text{ND}}$ - August, $30^{\text{TH}}$. This time

period is characterized by very active photo-chemistry in the Northern Hemisphere. The photolysis rate dictates a different kinetic environment for many key species of interest. To simplify interpretation, the analysis is performed on surface data (elevation = 1) and one latitude at a time, and for all 72 longitudes with data shifted in time as described above.

    In most of the latitudes in the Southern Hemisphere, the days are much shorter than the nights, and accordingly the daylight chemistry period is much shorter as compared to the nighttime chemistry period. Thus, the data exhibits a spiky pattern that

needs much higher modes to accurately reconstruct it; and/or we would need to isolate the day time values only when there are active chemical kinetics present. Hence, we are picking latitude = $30°$N for the analysis, which has the longest day times for the latitudes considered. The first 40 days of data is used as 'training' data, and the DMD diagnostics below are presented for this time period and for latitude = $30°$. With 72 snapshots per day we have a data matrix of $72(lon) \times 2880(time)$ for each latitude.





The optDMD is performed for this data matrix. We perform the analysis for six different chemical species of interest (Velegar
et al., 2019): Nitrous Oxide **NO**, Ozone **O₃**, Nitrous dioxide **NO₂**, Hydroxyl radical **OH**, Isoprene **ISOP**, and Carbon
Monoxide **CO**. For each species, we have **CONC** or absolute concentration data (expressed in units of molecules/cm$^3$) and
**TEND** or tendency/rate of change data (expressed in units of molecules/cm$^3$/s). Using the diagnostics from the 40 day training
period (July 2 - August 10), we then forecast the chemical evolution for the following 20 days (August 11 - 30).

### 3.1 DMD Diagnostics

The optDMD decomposes data into time dynamics represented by the spectrum of eigenvalues $\Omega$ and the corresponding
spatial modes $\Phi$. We will be presenting diagnostics from four different DMD approaches: (i) optDMD without constraining
the eigenvalues; (ii) optDMD with eigenvalues constrained to the left-half plane; (iii) optDMD with eigenvalues constrained
to the imaginary axis; and finally (iv) exact DMD. This is to examine which decomposition is best suited for reconstruction
and forecasting of the chemistry dynamics. The diagnostics are presented for the 40-day time series of the hydroxyl radical
species (**OH**). The results are consistent for all chemical species of interest. We have used a hard rank threshold truncation
of $r = 25$ for the **CONC** data and $r = 50$ for the **TEND** data. Truncating the rank for the DMD models is described below.
The diagnostics are presented for both absolute concentration of the chemical species, or **OH$_{\mathbf{CONC}}$** data, on the left panels
and rate of change of concentrations/tendencies due to chemistry, or **OH$_{\mathbf{TEND}}$** data, on the right panels in Fig. 5 and Fig. 6.
Four different spectra of the DMD eigenvalues are presented in Fig. 5, and the corresponding reconstruction of data is shown
in panels 2-5 of Fig. 6. The top two panels in Fig.6 are the actual **OH$_{\mathbf{CONC}}$** data on the left and actual **OH$_{\mathbf{TEND}}$** data on the
right, presented for comparison.

(i) The spectrum for optDMD with no constraints on the eigenvalues for **OH$_{\mathbf{CONC}}$** data is presented on the top left panel,
and for **OH$_{\mathbf{TEND}}$** data is presented on the top right panel of Fig. 5. For both data sets, some eigenvalues fall on the right-
half plane with positive real parts, causing the corresponding modes to grow in time. The corresponding reconstruction
of data is presented in the second two panels of Fig. 6. optDMD with no constraints does a faithful reconstruction of data,
but the forecasting results are poor, with the time series growing exponentially as a result of some eigenvalues on the
right-half plane. This approach is not used henceforth.

(ii) The optDMD is then constrained to produce only eigenvalues with negative or zero real parts, i.e. eigenvalues on the
closed left-half plane ($\Re(\omega_i \leq 0)$). The resulting spectrum for the two data sets is presented on the second two panels
in Fig. 5. The corresponding reconstruction of data is presented in the third two panels of Fig. 6. optDMD with these
constraints not only faithfully reconstructs the data, but the forecasting results are also accurate, as presented in the
following section.



**Figure 5.** *Comparing the spectrum for* 40 *day reconstruction results for Classical and Optimized DMD at the surface of* **OH** *preprocessed data. On the left 4 panels are the eigenvalues of* **OH**$_{\mathbf{CONC}}$ *data; on the right 4 panels are the eigenvalues of* **OH**$_{\mathbf{TEND}}$ *at* $\mathrm{Lat} = 30°$. *The top panels show the spectrum from Optimized* DMD *with no constraints, the second set of panels show the spectrum from Optimized* DMD *with linearized constraints that the eigenvalues be on the left-half plane, the third set of panels show the spectrum from Optimized* DMD *with linearized constraints that the eigenvalues be imaginary, and the bottom panels show the spectrum from Classical or Exact* DMD.

240   (iii) The optDMD is then constrained to produce only imaginary eigenvalues with zero real parts ($\Re(\omega_i = 0)$). The resulting spectrum for the two data sets is presented on the third two panels in Fig. 5. The corresponding reconstruction of data is presented in the fourth two panels of Fig. 6. optDMD with these constraints is not able to capture the data dynamics, and will not be used henceforth.







**Figure 6.** *Comparing* 40 *day reconstruction results for Classical, optimized DMD, and optimized DMD with no constraints at the surface of* **OH** *preprocessed data at* Lat = 30°. *The left panel is for absolute concentration or* CONC *data and the right panel is for* Tendency *data; the top panels show the preprocessed data, the second panels show the reconstruction from optimized* DMD, *the third panels show the reconstruction from optimized* DMD *with eigenvalues constrained to the Left half-plane, the fourth panels show the reconstruction from optimized* DMD *with eigenvalues constrained to the Imaginary axis, and the bottom panels show the reconstruction from the Classic* DMD. *The Classical* DMD *is unable to reconstruct the dynamics for the absolute concentration and tendency data.*

(iv) Finally, results from Exact DMD for both data sets are presented in the bottom two panels of Fig. 5 and Fig. 6. The resulting spectrum for the two data sets have most eigenvalues on the negative real axis, implying decaying modes. The corresponding reconstruction of data also decays out with no dynamics from the data captured or represented faithfully. This approach is not used henceforth.





Thus, we will use optDMD with eigenvalues constrained on the closed left-half plane $\Re(\omega_i \leq 0)$. When computing the opt-DMD, we truncate the number of modes to avoid fitting dynamics to the lowest energy modes, which may cause over-fitting and may be corrupted by noise. We would be truncating using *hard-thresholding* at a rank $r$ at which the relative error in reconstruction has an 'elbow', i.e. the error graph flattens out without further decrease. Focusing on six key chemicals of interest: **NO**, $\mathbf{O_3}$, $\mathbf{NO_2}$, **OH**, **ISOP**, **CO**, **CONC** and **TEND** data, we now compute the relative error as we increase the number of modes from 1 to 50. The results for the two data sets and the six chemical species is presented in Fig. 7. A larger number of modes is needed to reconstruct the **TEND** data as compared to the **CONC** data. Based on the results, we use 20-30 modes for optimal diagnostics of **CONC** data, depending on the chemical species. For the **TEND** data we pick between 30-50 modes.

Finally, we present the global spatial modes for **CO** and **NO** computed at 12° latitudes -14° through 30° in Fig. 8 and Fig. 9 respectively. The 12 latitudes are selected for having consistent day lengths across all longitudes and at least 4 snapshots during day time. As described above, the optDMD is performed for one latitude at a time to have consistent day time lengths across all the time series, and the resulting spatial modes are pieced together to present a global picture. The underlying spatial features of the data sets are resolved well by the constrained optDMD diagnostics. The high-variance features at the coastlines and within hot spots in the land for the chemical species are represented clearly.

## 3.2 Forecasting

As described above, using an appropriate rank truncation, the optDMD with eigenvalues constrained to the closed left-half plane faithfully reconstructs the time series data for 40-day training window and a given elevation/latitude. We now forecast the time series data for future times beyond the training window. Using (1), with amplitudes **b**/modes $\mathbf{\Phi}$/eigenvalues $\mathbf{\Omega}$ computed by optDMD during the training window, we forecast time series for the subsequent 20 days. The results for **CONC** and **TEND** data for two chemical species **OH** and **NO** are presented for 6 longitudes, and latitude 30° at the surface(elevation=1) in Figures 10, 11, 12, and 13.

Constrained optDMD faithfully reconstructs and forecasts the time series for the 20 days tested. Since we use the fewest modes possible, spikes in actual data are sometimes not reproduced and we see a sinusoidal best fit time series instead. The $\mathbf{NO_{TEND}}$ results in Fig. 13 demonstrates this.



**Figure 7.** *Relative Error plotted against number of modes used for Optimized* **DMD** *with eigenvalues constrained to the left-half plane; for 6 different chemical species and CONC and Tend data at Latitude=30°*

We have snapshots of the data every 20-minutes, hence 72 snapshots per day. We compute the relative error for all longitudes for each day, and average across space and snapshots for each day. The resulting mean relative errors are presented for all 6 chemical species of interest and for both **CONC** and **TEND** data in Fig. 14 in color red. The 95-percentile confidence intervals for each day is presented as black bars, indicating the variance for the mean relative errors. Constrained optDMD does an excellent job in forecasting the immediate future snapshots and does consistently well during the entire 20-day data tested, with mean errors/uncertainty in forecasting increasing only slightly for some chemical species as the number of prediction days increases away from the last snapshot used from training. No exponential growth/decay is observed in the forecast time-



**Figure 8.** 40 *day reconstruction results for Optimized DMD at the surface of* **CO** *preprocessed data. The analysis was computed for 12 latitudes -14° through 30°. The left panel show the dominant four spatial modes for CONC data; and the right panel show four of the corresponding spatial modes for the TEND data.*

series, while the underlying dynamics are forecast faithfully. Considering that the underlying dynamics represent a moving state with time, the constrained optDMD minimizes model bias with the variable projection optimization, thus leading to stable forecasting capabilities. The performance is slightly worse in forecasting the **TEND** data as compared to the **CONC** data, which is due to the intrinsic rank of the **TEND** data being higher. Increasing the truncation rank of the projection would lead to improvement in forecasting of the **TEND** data.



**Figure 9.** *40 day reconstruction results for Optimized DMD at the surface of* **NO** *preprocessed data. The analysis was computed for 12 latitudes -14° through 30°. The left panel show four spatial modes for CONC data; and the right panel show four of the corresponding spatial modes for the TEND data.*

The optDMD performs worst in forecasting the chemical species **OH**. OH has a very short tropospheric lifetime of less than a second and exhibits rapid chemical cycling during the daytime. Consequently, this chemical species needs the highest number of modes to capture its dynamics (Fig. 7).

### 3.3 Temporal Uncertainty Quantification



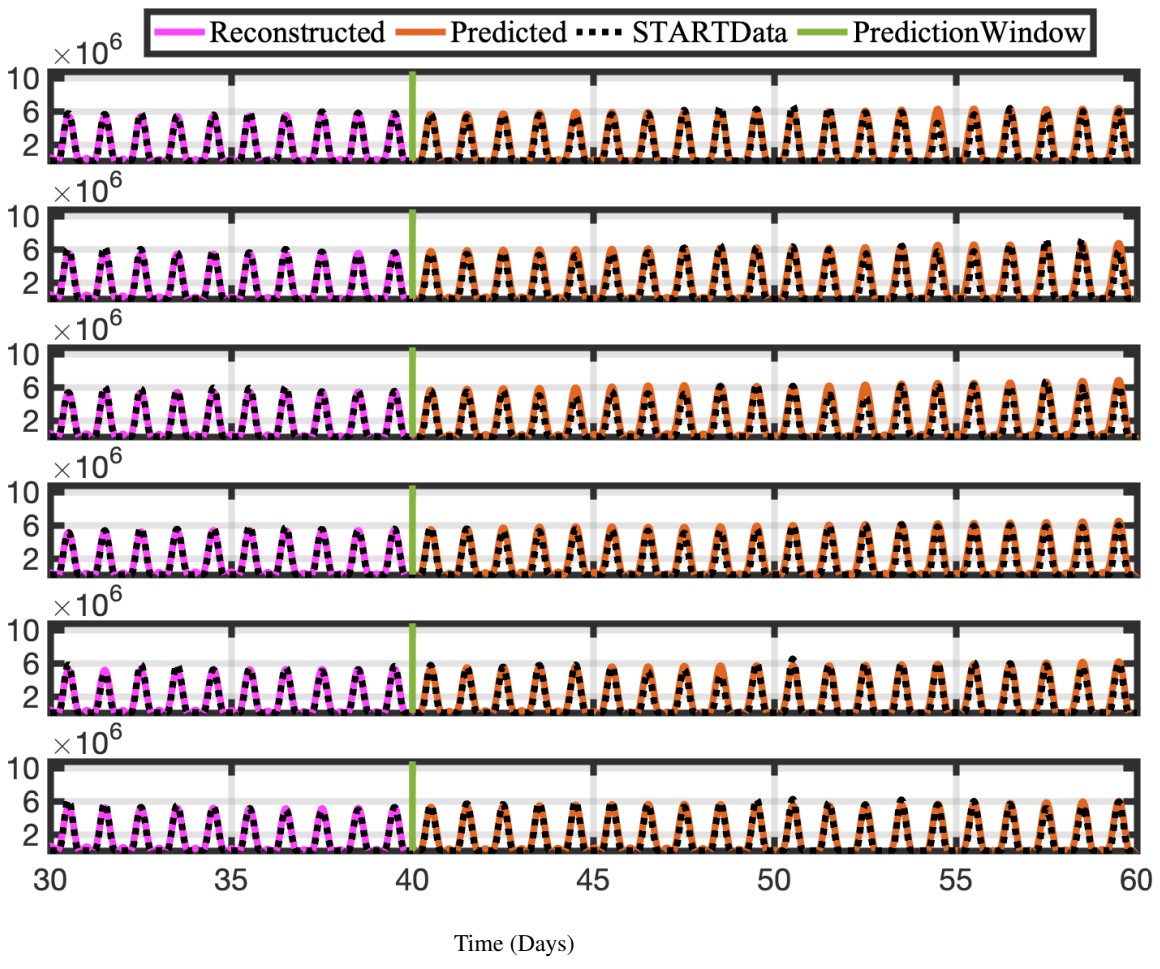

**Figure 10.** *Time series of reconstructed and predicted results with* $\mathbf{OH_{CONC}}$ *data at Lat 30° and 6 longitudes -180°:5°:-155°. Both the reconstructed data, shown here for 10 days; and the forecasted time series, shown here for the 20 day testing period, faithfully reconstruct and forecast the actual data for* $\mathbf{OH_{CONC}}$.

We now present the results from BOP-DMD in partnership with the optimized DMD algorithm to produce ensemble models and compute temporal uncertainty for the eigenvalue spectrum of both **CONC** and **TEND** data for the six chemical species of interest at Lat 30°. We use the constrained optDMD as described above on a full training data set of 60 days (July, 2$^{\text{ND}}$ - August, 30$^{\text{TH}}$) to create an initial seed $\mathbf{\Phi}_0, \mathbf{\Omega}_0, \mathbf{b}_0$ for the BOP-DMD algorithm . For $K = 100$ trials, we randomly select $p = 216$ snapshots/columns i.e. data for 3 days out of the 60 days to create our subset of data, as shown in Fig. 4. optDMD now computes the eigenvalues of various subsets using the aforementioned initial conditions. The $K = 100$ ensemble models' eigenvalues are used to produce the temporal UQ metrics.



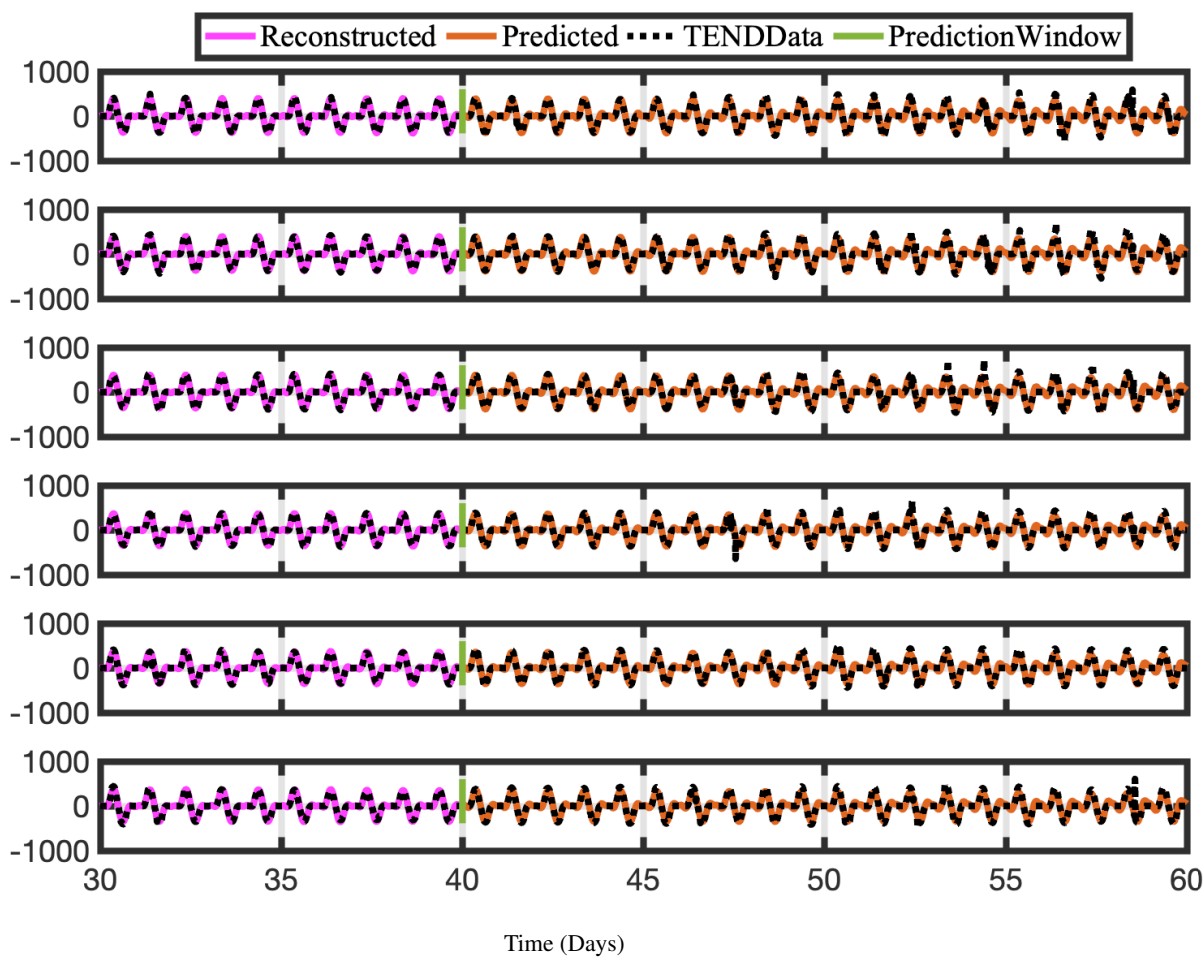

**Figure 11.** *Time series of reconstructed and predicted results with* $\mathbf{OH_{TEND}}$ *data at Lat 30° and 6 longitudes -180°:5°:-155°. Again, both the reconstructed data, shown here for 10 days; and the forecasted time series, shown here for the 20 day testing period, faithfully reconstruct and forecast the actual data for* $\mathbf{OH_{TEND}}$*.*

Fig. 15 shows the BOP-DMD distributions of the absolute value of the first five eigenvalues for each of the subsets of data for $\mathbf{OH_{CONC}}$ and $\mathbf{OH_{TEND}}$ data at Lat 30°. The BOP-DMD quantifies the temporal uncertainly by allowing for a Gaussian fit, shown in red. For both of the data sets, we see a high temporal uncertainty in eigenvalues with outliers skewing the distributions. The temporal uncertainty gets worse for the higher modes in the $\mathbf{OH_{CONC}}$ data and for all modes of $\mathbf{OH_{TEND}}$ data. Then we trim the eigenvalue distribution data to exclude the outliers below 10-*percentile* and above 90-textitpercentile to improve

the UQ metrics. Fig. 16 shows the distributions of the trimmed absolute eigenvalues, and the Gaussian fit is better with lower



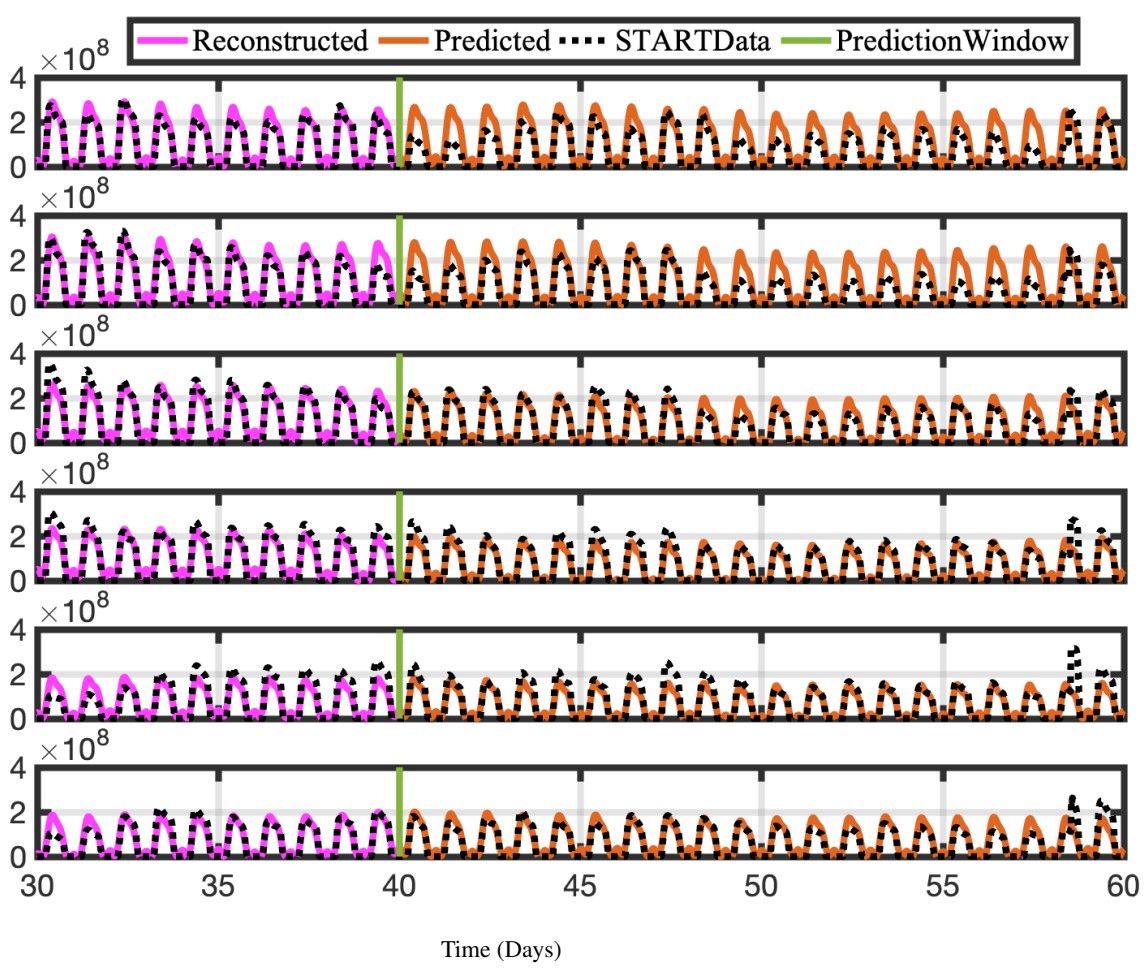

**Figure 12.** *Time series of reconstructed and predicted results with* $NO_{CONC}$ *data at Lat 30° and 6 longitudes -180°:5°:-155°. Both the reconstructed data, shown here for 10 days; and the forecasted time series, shown here for the 20 day testing period, reproduce the actual data for* $NO_{CONC}$ *well.*

variances, and only 1 distribution with outliers. Still, we see that there is significant temporal variability, especially for higher modes for $OH_{TEND}$.

## 4 Discussion

Based on the results presented in this work, we conclude that the constrained optDMD is the DMD algorithm of choice for
the reconstruction and forecasting of global atmospheric data. Exact DMD fails in the task of reconstructing the chemistry



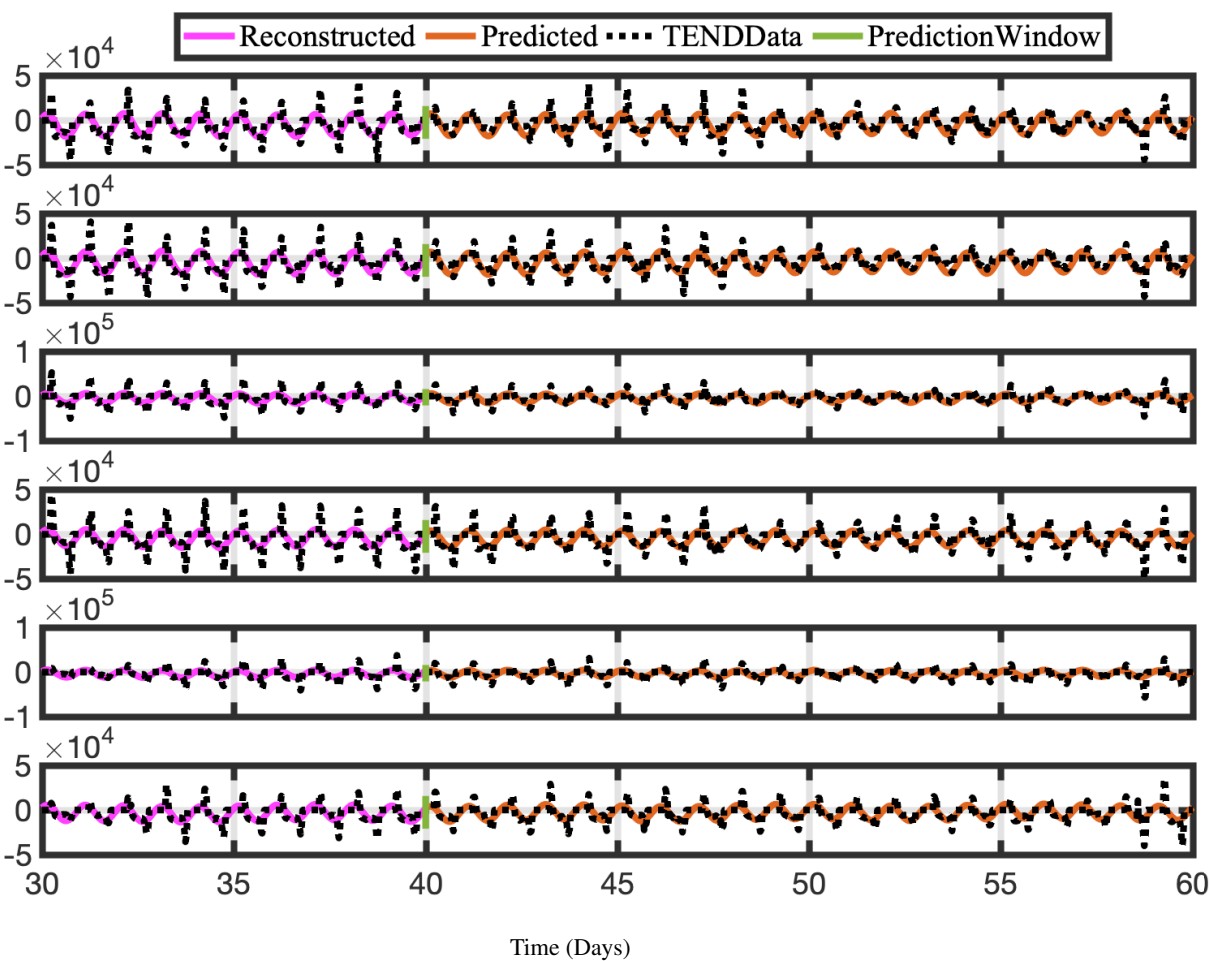

**Figure 13.** *Time series of reconstructed and predicted results with* $\mathbf{NO_{TEND}}$ *data at Lat 30° and 6 longitudes -180°:5°:-155°. Both the reconstructed data, shown here for 10 days; and the forecasted time series, shown here for the 20 day testing period, do not capture the spikes in the actual data for* $\mathbf{NO_{TEND}}$. *Since we are using only 20-30 modes for reconstruction, we get a sinusoidal best fit.*

time-series it is regressed to, let alone producing a reasonable forecast. This is due to the significant bias in the model from energetic localized convective phenomena present in the atmospheric simulation data. The optDMD algorithm casts the regression problem as a nonlinear optimization enabled by variable projection techniques (Askham and Kutz, 2018), hence providing an optimal de-biasing for the atmospheric chemistry dynamics. The optDMD is thus better able to capture hidden dynamics, showing an order of magnitude improvement in the reconstruction error. optDMD also produces modes which more accurately describe the localized energetic convective phenomena in the **CONC** and especially the **TEND** chemistry dynamics. The nonlinear optimization problem in the optDMD also allows for constraints. By adding a constraint $\Re(\omega_i \leq 0)$ to the optDMD



**Figure 14.** *Mean relative error with 95-percentile confidence intervals forecasting* **CONC** *and* **TEND** *data at Lat 30° for a prediction window of 20 days; and for 6 different chemical species. The relative error stays nearly the same or changes only slightly as the number of days we are forecasting out to increase.* **optDMD** *does better at forecasting* **CONC** *data as compared to the* **TEND** *data.*

minimization, we obtain accurate eigenvalues that are able to produce high-fidelity stable and robust forecasts. For the entire testing time window, the forecasts remain accurate as we increase time away from the training time window, not displaying any

growth, decay or loss of accuracy. However, computing the optDMD requires solution of a nonlinear, nonconvex optimization problem, which often fails to converge to a solution. The computational cost of the optDMD is higher, as we increase the number of snapshots, the cost increase becomes more significant. The solutions obtained here nevertheless represent significant improvements. Partnering the optDMD algorithm with the statistical bagging and ensembling of the BOP-DMD produces tem-



**Figure 15.** *Temporal uncertainty quantification for absolute of eigenvalues for* $\mathbf{OH}_{\mathrm{CONC}}$ *and* $\mathbf{OH}_{\mathrm{TEND}}$ *data at Lat* $30°$. *The red lines represent a least-square fit of a normal distribution. 60 days of training data was used with a sample size of 3 days and 100 cycles.*

poral UQ metrics, and highlights the high temporal variance in the eigenvalues produced by optDMD. This temporal variance

gets worse for higher modes of the **CONC** data; eigenvalues for the **TEND** data have quite high temporal variance.

An interesting further direction would be to apply the optDMD to an entire year's worth of data, a still computationally tractable problem. In particular, the current study did not look at the ability of optDMD to faithfully reproduce yearly patterns in the chemistry data, and accurately forecast seasonal variations. The BOP-DMD can be leveraged to produce spatial UQ metrics, illustrating the spatial patterns where optDMD is most uncertain in it's ability to provide accurate representations. optDMD

can be further empowered by partnering with the BOP-DMD by (i) an initialization procedure to stabilize it's convergence, improving the robustness and accuracy of the regression, (ii) leveraging statistical bagging to produce a stable model with



**Figure 16.** *Temporal uncertainty quantification for absolute of trimmed eigenvalues for with* **OH**$_{\mathrm{CONC}}$ *and* **OH**$_{\mathrm{TEND}}$ *data at Lat 30°. The data has been trimmed to remove outliers below 10 percentile and above 90 percentile. The red lines represent a least-square fit of a normal distribution.*

reduced variance in the model parameters, and (iii) leveraging this stable model to forecast future states of spatio-temporal atmospheric chemistry system, with Monte Carlo simulations to produce UQ for future states.

The here presented approaches have the potential to produce reliable estimates of 'business-as-usual' patterns of global
atmospheric composition in real-time and at very low computational cost. They are not designed to capture unusual events such as air pollution due to wildfires or sudden pollutant emissions changes (as e.g. experienced in the wake of the COVID-19 outbreak). However, when combined with actual atmospheric observations, the presented method can be used to identify and quantify air pollution anomalies.





**Author Contributions:** Conceptualization, J.N.K. and M.V.; methodology, M.V. and J.N.K.; software, M.V.; validation, M.V., C.K. and J.N.K.; formal analysis, M.V., C.K. and J.N.K; resources, C.K. and J.N.K.; data curation, C.K. and M.V.; writing—original draft preparation, M.V., C.K. and J.N.K.; writing—review and editing, M.V., C.K. and J.N.K.; visualization, M.V.; supervision, J.N.K. and C.K.; funding acquisition, J.N.K. . All authors have read and agreed to the published version of the manuscript.

**Funding:** The authors acknowledge support from the National Science Foundation AI Institute in Dynamic Systems (grant number 2112085). JNK further acknowledges support from the Air Force Office of Scientific Research (FA9550-19-1-0011).

*Data availability.* The code is openly available on the following github link https://github.com/mvelegar/DMDPaper. The code and data area available on zenodo: 10.5281/zenodo.12754943.

**Conflicts of Interest:** The authors declare no conflict of interest. The funders had no role in the design of the study; in the collection, analyses, or interpretation of data; in the writing of the manuscript; or in the decision to publish the results.





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
