# Peer review of "Optimized Dynamic Mode Decomposition for Reconstruction and Forecasting of Atmospheric Chemistry Data"

_Geoscientific Model Development, 2024_

## Author Response (AR2)

Dear Editor

RE: GeosChem Rebuttal

Please find attached the original rebuttal document.  The original referee comments were in **black**, with our responses in red and the manuscript changes in blue.  We have not added additional changes to the manuscript in **magenta** as requested by the editor.

Apologies for not having highlighted these earlier in the manuscript.

Sincerely

J. Nathan Kutz
Corresponding author

Additional Editor Comments

Thank you very much for providing detailed answers to the referees questions and suggestions, and sorry for the delay in the editorial process. Although you, from my point of view, answered all concerns raised by the two referees adequately, this is not all reflected in your revised manuscript.  Please note that the final manuscript needs to be self-explanatory, and therefore it is not sufficient to answer the question in the reply, in most of the cases, adaptions of the manuscript are required as well (to later not trigger the same questions by the reader). Therefore I kindly ask you for another revision for which you consider textual changes / additional explanatory text. In particular:

Thank you for your comments, below you will find explicitly were all additional comments and responses have been made in the manuscript.  The new **magenta** text shows the additional comments made in response to the comments.

According to referee #1 (the original comments):
- "6. Line 123 what does elevation = 1 mean physically? Are these units here?": This is answered in the reply, but in the revised text it remains unclear. I suggest to write "elevation=surface" instead, the same where else this occurs in the text.

Apologies.  We have now corrected this and the following is now in the manuscript at linear 140 when elevation=1 is first introduced

"We further note that out of the large number latitude, longitude and elevation settings, we highlighted surface dynamics (elevation $=1$) as this elevation is not only rich dynamically,

but it is also the elevation on which humans are exposed to the atmospheric chemistry dynamics. As will discussed in what follows, we have made judicious choices to demonstrate the dynamics present."

- 13: "Lines 254-255 can you specify exactly how many modes were used for each method in a table?": Please add information to the revised text (as in your reply) where appropriate.

The rank (modes) of the decomposition was highlighted in the text, but has been further clarified in Figs. 5 and 6 to be explicit.

"Note that a hard rank threshold truncation of $r=25$ for the \textbf{CONC} data and $r=50$ for the \textbf{TEND} data has been used."

- 16: "Figures 15,16 what does i,j represent in $\langle _{i,j}^2 \rangle$?": Please add this information to the figure captions.

This has been fixed with the following addition to the caption

"The complex conjugate pair frequencies are denoted by $\langle \bf \omega_{i,j}^2 \rangle$ where for the pairing $j=i+1$. Thus $\bf \omega_1$ and $\bf \omega_{2}$ are the complex conjugate pairs whose variance is evaluated jointly."

- 19: "Figure 8, 9 what do the subscripts i,j in $\phi_{i,j}$ represent?": Please add this information to the figure captions.

This has been fixed with the following addition to the caption

"The complex conjugate pair of DMD modes are denoted by $\boldsymbol{{\phi_{i,j}}}$ where for the pairing $j=i+1$. Thus $\bf \omega_1$ and $\bf \omega_{2}$ are the complex conjugate pairs whose real parts are identical."

- 20: "Figures 10-13 does it make more sense to plot the MSE and correlation between STARTData and the OptDMD reconstructions/predictions at each time step rather than the raw signals? By doing this, one could compress these four figures into one or two figures for easier comparison of the results across different chemical species and CONC vs TEND predictions.": Your answer is "We can certainly do this and can get this done fairly soon to put into the manuscript.", but there is not change in the revised manuscript. Please go ahead and apply this change.

Thank you for the comment. We had quite a long conversation internally about this and how to proceed. We actually think that Fig. 14 is much more informative as it shows the error across an entire forecast in time. So instead of just one score, one case not only see the mean and variance of the error, but also how the error evolves in the forecast. We think this is a much more informative way to understand the error for forecasting with DMD.

According to referre #2 (the original comments):
- "The Authors propose interpretability as an additional advantage (In abstract). This should be elaborated (and
compared with other modeling approaches) in the discussions.": please add your reply as is or reformulated to the revised text, where appropricate

- "l.105: Model output was for 1 year. Why later on about 40/60 days' data have been used for training and validation, to save computation time? How does length of data impact the performance?": please add corresponding information from your reply to the text, where apppropriate

This has been addressed with the following text changes

"The number of days used for fitting (40 days) is one of two hyper-parameters for the DMD regression, the other being the number of modes (rank) used. A sliding window approach for sampling for DMD has been shown to be quite effective for reconstruction and forecasting~\cite{kutz2016multiresolution,lapo2024unsupervised}. Typically a shorter sampling window helps in forecasting as the often data is non-stationary and long time histories compromise the DMD model. Thus we use a fairly models history of 40 days for forecasting, which also makes the model smaller to manage. In general, this is also in keeping with the DMD philosophy of a model that can be simply run again due to its small computational footprint."

- "l.112: nfeatures = 143 + 91 + 3 + 143 = 380; mention what each number is; (143 is given to be chemical species, what about other numbers here)":
In the revised manuscript, "nfeatures" occurs only once on line 119, but it is not at all exlained. Please do so.

This has now been amended with the following additional text

"The 380 in the feature space breaks down as $143 + 91 + 3 + 143 = 380$ which refers to the chemical species concentration before integration, the photolysis rates, the 3 meteorological variables, and the tendencies (rate of change) of all species due to chemistry as specified in the GEOS-Chem simulations \url{https://geoschem.github.io}."

- "l.220: "The results are consistent for all chemical species". What exactly this statement means? [...]": Please add the information from your reply to the revised text.

This has been clarified as follows in the text:

"Specifically, the forecasting performance and error is agnostic to the specific chemical species considered, thus suggesting the DMD behavior is independent of the specific chemistry being modeled."

- "l220: [...] Describe rationale behind choosing ranks (r=25 for CONC and r=50 for TEND)?": Please do so in the revised text.

This has been clarified as follows in the text:

"These specific target ranks are chosen through hyper parameter tuning of their forecasting performance. Too few modes compromises the DMD model since there are not enough features to accurately reconstruct and forecast the data. Too many modes overfit on the training data. So although arbitrary, these specific values show generically strong performance across chemical species for the task of forecasting."

- "l.276-278: missing spikes are attributed to selection of fewest modes possible. What is the rationale behind not taking few more modes and trying to improve model performance. Air quality spikes are often of significant interests.": Please add the important information from your reply to the revised text, where appropriate.

The following has been added to Fig. 13. This is a really hard task for any time-series method unfortunately.

"In general, spikes in time-series data are difficult to capture and forecast with any method, including with DMD. Although more modes can provide a better reconstruction, it often is then overfit on training data for forecasting purposes."

- "Fig. 10: the legend STARTData does not seem to be defined in manuscript. Please check.": Despite your reply, the figure in the revised manuscrip is unchanged.

Apologies, this should be CONC, not START. So we have updated this in the figure.

- "If possible do include results on ozone (O3) also, similar to that done for CO, NO (Fig 8-9). Ozone is affected by photochemistry as well as transport (and is precursor of OH). Getting O3 distributions right could be very useful.": Despite your reply, the revised manuscript does not contain a figure of ozone.

The O3 data is quite erratic and intermittent. We include a figure here to illustrate the behavior. Such intermittent behavior is difficult for any time-series method to capture, including DMD.

"There are, of course, limitations in the methodology presented, especially when considering chemical dynamics that are highly intermittent and which lack any periodic, or quasi-periodic behavior. Ozone is an example of a chemical which is intermittently active in its dynamics, thus compromising the ability of an algorithm like DMD to produce quality reconstructions and forecasts. Such chemical have been excluded from consideration as methods for such time-series behavior are currently lacking."

Please revise your manuscript accordingly and provide a document on which (this time all!) changes compared to the first revision are highlighted.

We have modified the manuscript as requested and include all the new changes in magenta in order to highlight both the previous and current revisions.

Thank you for the careful reading which has allowed for us to more carefully explain variables, workflow and calibration.

Referee #1

It is always exciting to see DMD variants applied to datasets in earth systems science. This paper considers one such application and sets the stage for more detailed analyses of such datasets using OptDMD and BOPDMD. The authors begin with a small example of atmospheric chemistry data that emphasizes the superiority of OptDMD over exact DMD. Then, OptDMD is used to reconstruct and forecast atmospheric chemistry data and BOP-DMD is used to provide uncertainty quantifications and other forecasts. These forecasts, along with their errors appear stable over the prediction windows. They also use the dynamic modes detected from OptDMD to visualize the spatial modes of these data.

One of my general concerns is a lack of scientific interpretation of the results. Some of my questions include: why choose only certain chemical species, latitudes, longitudes, and elevations for these experiments?

These are valid and constructive questions. The data set we used is quite extensive with hundreds of chemical species and with a large global grid of lat, long and elev values. We had to make strategic choices about what to present. As such, we relied on the domain expertise of one of our PIs (Keller, who unfortunately is no longer at NASA) for selecting commonly

considered chemicals.  Specifically, out of the hundreds of chemicals for us to show, the dynamics of Nitrous Oxide ${\mathbf{NO}}$, Ozone ${\mathbf{O_3}}$, Nitrous dioxide ${\mathbf{NO_2}}$, Hydroxyl radical ${\mathbf{OH}}$, Isoprene ${\mathbf{ISOP}}$, and Carbon Monoxide ${\mathbf{CO}}$ are commonly considered in the literature and as a diagnostic.  We have made the code open source so that with laptop level computing, DMD can be performed on any specific chemical of interest.  We realize that these choices are somewhat arbitrary, however, they do help to represent the underlying atmospheric chemistry dynamics along with the ability of the DMD regression to predict and reconstruct the evolution.

Thus the same diagnosis can be applied to any of the other chemicals. We have further explained in the paper that we had to pick latitudes that had a significant day length since the dynamics have much less variability due to lack of sunlight. For most of the results we have used Latitude 30 since it has consistently the longest day so there is a lot of variability in the dynamics to analyze. All longitudes are studied, but since we could not present figures for all of them, we picked 6 longitudes in the east. The same is true for elevations, the diagnosis can be applied to any elevation so we chose surface level for all the chemicals. I have checked all longitudes and tried different elevations and the results were consistent, i.e., I could run the diagnosis and interpret the results in the same way.  Again, the number of figures needed to represent a comprehensive set of chemicals, longitudes, latitude, and elevations becomes quickly intractable and requires a selective downsampling of important values, which we feel we have done.

The following text has been added just before Sec. 3.1
"Although there are hundreds of chemicals whose dynamics can be demonstrated, the six selected are chemicals commonly associated with atmospheric diagnostics, including pollution and environmental health.  Similarly, out of the large number latitude, longitude and elevation settings, we highlighted surface dynamics as these are often some of the richest and most relevant for understanding the role of atmospheric chemistry affecting humans.  It is an intractable task to show all chemicals at all locations.   Thus the judicious choices represent those of greatest impact and which are commonly considered by experts in practice.  The code provided allows one to consider any chemical at any location desired."

Can you provide a more in-depth interpretation of the spatial modes?

Unfortunately, our lead domain expert (Keller) left NASA upon completion and submission of the work.  He certainly had the capability of going into great detail on the modal structures and how they represent well-known spatio-temporal trends in the community.  Regardless, the paper is much more focused on the algorithm's ability to produce the these patterns and forecast the future states for different chemicals at different lat, long, elev positions.  As such, the algorithm accomplishes the primary task of the paper.  We have added the following at the beginning of Sec. 2

"Many of the dominant spatio-temporal features of atmospheric chemistry are well-understood through extensive simulation and data

collection~\citep{jacob1999introduction,brasseur2017modeling}. This will not be the focus of this work, but rather a robust, computationally efficient and accurate reduced order model for reconstructing and forecasting the dynamics."

Further comments are below:

On lines 6-7 authors claim that their method "successfully extracts known major features of atmospheric chemistry, such as summertime surface pollution and biomass burning activities." But, the only other time "pollution" is mentioned is on lines 339-343, where the authors state that these methods have "the potential to produce a reliable estimate of business-as-usual patterns." I may have missed something, but I couldn't find something in the results where the authors specifically point out the detection of summertime surface pollution and biomass-burning activities.

Apologies, we have removed this from the abstract. Although the method can do this, we have not really spent time with the interpretation, but rather the implementation and forecasting capabilities. Thus the abstract has been modified as follows

"We show that the presented decomposition method successfully extracts and forecasts chemical patterns for leading chemical indicators, including nitrous oxide, ozone, nitrous dioxide, hydroxyl radical, isoprene, and carbon monoxide."

Line 59 "highlighted" should be "highlights"

Thank you. We have fixed this.

Can the authors elaborate on why they choose to only consider eigenvalues with a non-positive real part other than the fact that it produces "accurate eigenvalues and high-fidelity stable and robust forecasts?" Does this choice to restrict the eigenvalues to align with some physical assumption in atmospheric chemistry?

Thank you for the question. One of the weaknesses of standard DMD without constraints is that the regression process will generically produce eigenvalues which often have real positive part. In fact, in early DMD work, the eigenvalues can often be "pushed around" by noise and create artificial decay or growth, which is very nicely overcome by optimized DMD. In fact, optimized DMD allows for constraining the eigenvalues so that forecasts don't blow up to infinity. A positive real part of an eigenvalue ensures that even a moderate future forecast starts to blow up exponentially. The restriction guarantees stable "roll outs" or forecasts by design, allowing for long-time forecasts without the detrimental effects of artificially induced exponential explosion.

The following has been added to the text in the DMD diagnostics sub-section

"The constraints are important in practice, especially for forecasting the atmospheric chemistry. Without constraints, and often due to noise, the data can generate eigenvalues which have positive real parts. Even moderate length forecasts will blow up artificially due to the real part being positive. The optDMD algorithm allows us to remove this unbounded artificial exponential growth. Growth of the solution is still accommodated by modeling it as the first part of an oscillatory solution (which looks like it is growing, but which is in reality an oscillating mode). Similarly, it has already been noted that noise can also artificially bias the eigenvalues towards the left half plane which makes solutions decay to zero. Thus a forecast will exponentially die away to zero. The constraint of eigenvalue on the imaginary axis guarantees a stable long-term forecast that neither grows nor decays. Of course, this is a pure regression problem which induces its own limitations, but in regards to forecasting, it has the important and desirable properties of stability for long-term forecasting. There is an additional inherent assumption with constraining the eigenvalues to the imaginary axis: conservation of mass of that chemical species."

Figure 1. Should $x_m$ be bold-face? Can you be more specific in Figure 1 with what exactly "the data $\mathbf{x}(t_k)$" represents in terms of the global chemistry discretization? I imagine this changes per experiment, but maybe you can be more specific.

Yes, $x_m$ and $x'_m$ should be bold-face and this has been fixed. Below equation 1 I have defined what $\mathbf{x}(t_k)$ is, the only discretization is stacked longitude cells, with the same latitude and elevation.

Line 87 shouldn't "unpractical" be "impractical"?

Thank you, this has been fixed.

Line 123 what does elevation = 1 mean physically? Are these units here?

It's the surface level elevation. The elevation grid is not linear, the elevations in the discretization are crowded closer together near the surface and become more spread out at higher elevations.

Lines 130-133, 207, 251, 339, please use ` for the left quotation mark

Thank you, this has been fixed.

Lines 133-134 Should the sentence "On the bottom right … " belong in the figure caption

Thank you, this has been fixed.

Figure 2- Why did the authors choose latitude 30?

Please see above.  You could say this is just a judicious choice.  We wanted to pick a latitude of the equator, and 30 degrees is commonly looked at.

A space is missing between the meridian and (Lon=...

Thank you, this has been fixed.

Lines 160-163 is there any physical interpretation why this restriction of eigenvalues makes sense? Could this have something to do with energy conservation?

Please see above for the implications:  namely that the chemical species concentration (mass) is preserved for eigenvalues on the imaginary axis.

Line 177 and 184 the reference to algorithm 1 is missing.

Thank you.  This has been fixed.

Figures 3, 5, 6, 10, 11, 12, 13, 15, 16   Please label the rows of the figures.

The figures looked too crowded/small for me to label the rows, hence I used the captions to explain what each row represents. Basically, we tried maximize the figure while having a common x-label for all of them.  Labeling each makes the figures even bigger and also reduces what we would like the reader to see better.

Lines 254-255 can you specify exactly how many modes were used for each method in a table?

This is variable based upon a prescribed cut-off.  Thus we used the range 20-30 modes. Although not provided currently, we can produce an additional table with these results before finalizing a draft.

Line 259 remove the degree sign from 12

Thank you.  This has been fixed.

Line 264-265 Why does it make sense to have "high-variance features at the coastlines and within hot spots in the land?" Has this been seen before in prior work? If so, please provide a citation.

We have added citations to some of the foundational work where such features are commonly known among the atmospheric chemistry community.

Figures 15,16 what does i,j represent in $\langle _{i,j}^2 \rangle$?

The order of the eigen values. i=1 and j=2 are the first and second most significant eigen values (with the same distributions since they are complex conjugates).

Line 269-271 why is the analysis restricted to only 6 longitudes, 1 latitude, and 1 elevation? Which longitude is used? Also, a space is missing between surface and (elevation=1). Again, what does elevation=1 mean physically? What are the units?

Please see above.  We simply made choices that were representative and commonly used in the community.   In the end, we had to make some judicious selections.

Figure 7,  14, 15, 16 shouldn't Tend be TEND

Thank you. This has been fixed.

Figure 8, 9 what do the subscripts i,j in $\phi_{i,j}$ represent? Also, shouldn't it be "shows" instead of "show?"

The order of the eigen values. i=1 and j=2 are the first and second most significant eigen values. Yes it should be "shows".

Figures 10-13 does it make more sense to plot the MSE and correlation between STARTData and the OptDMD reconstructions/predictions at each time step rather than the raw signals? By doing this, one could compress these four figures into one or two figures for easier comparison of the results across different chemical species and CONC vs TEND predictions.

We can certainly do this and can get this done fairly soon to put into the manuscript.

Figures 15-16 What is the advantage of plotting the uncertainty quantification of these eigenvalues? What does this tell us about the model and the resulting predictions? Does this provide us with any insight into the physical processes?

This is a great question.  What having the distributions of eigenvalues allows is for a probabilistic prediction of the future state.  Specifically, by drawing from the eigenvalue distribution, we can produce an ensemble of forecasts and compute the growing variance of this forecast as a function of the forecast horizon. This is a standard thing to do in time-series analysis, and it can be easily and simply be done with DMD, unlike many expensive Monte-Carlo type simulations.  This then allows us to understand the growing UQ in the forecast itself and in some sense, the Lyaponov exponent of the system, ie the time for which small perturbations make future state predictions diverge.  It is still quite an open question what this

means for the underlying physical processes, but it does give an important metric about the confidence of a prediction.

The following has been added to the UQ section:

"The UQ metrics are critical for understanding the ability of the BOP-DMD algorithm to perform long term forecasting.  Specifically, BOP-DMD is a low-cost computational tool, as opposed to Monte-Carlo simulations, for evaluating the divergence of future state predictions from an ensemble of predictions, specifically drawn from the BOP-DMD eigenvalue distribution."

There are certainly other type-os that I have missed. Please take a very careful pass through the manuscript to check for any other type-os before submitting revisions.

We are indebted to you for your careful reading.  It was tremendously helpful to fix up the manuscript.

Referee #2

Atmospheric chemistry simulations over large spatial regions are computationally expensive and the new developments of alternative modeling approaches, such as statistical and AI/ML are increasing. In this regard, Velagar et al. have presented a comprehensive study exploring the application of Dynamic Mode Decomposition for simulating spatio-temporal variability in chemical species. Out of several experiments, they suggest that optimized DMD with constraint can be a better model for atmospheric chemistry application. While the study is novel, detailed, and in-general well written, some comments need to be addressed before publication, as listed below.

The DMD approach is explored to reduce computation speed. If possible, provide some estimation on how much time this method is taking for the considered periods of training / simulations. How it is varying with complexity (number of chemical species, etc.). Authors propose interpretability as an additional advantage (In abstract). This should be elaborated (and compared with other modeling approaches) in the discussions. I am not quite sure how this approach has more interpretability than for example machine-learning approaches which also provide relative significance of different features in governing variability?

The DMD model is unlike machine learning methods which require training and cross validation. The DMD is a direct regression on the data which can be performed in seconds on a laptop. And with randomized algorithm, one can even perform the DMD decomposition in seconds on terabtyes of data.  We've added the following to the manurscript:

"The SVD/DMD can even be done on Terabytes of data in seconds~\cite{eiximeno2025pylom}."

As for interpretability, the DMD gives a mode and a frequency.  For spatial-temporal systems, this is, we would argue, the most interpretable model possible.  It simply says that a spatial pattern that is found has a certain oscillation frequency.   Moreover, the modes are a linear combination.  No machine learning model comes even close to such simple interpretability.  So we are not arguing against machine learning, especially as one of the authors (Kutz) works extensively in ML/AI, but rather, DMD is an incredibly efficient tool that one can do in seconds on simple compute platforms without training, cross-validation or hyper-parameter tuning.

l.15: 5-dimensional data! Would not it be better to call 4-D data, variability in chemical species are 4-D data, e.g., O3(lon, lat, alt, time)?

Thank you.  We have made the requested change.

l.52-55: "these limitations make their use in global atmospheric modeling problematic.". The machine learning approaches for atmospheric modeling are still in developmental stages with several successful applications (e.g., Arcomano et al., GRL, 2020; Kochkov et al., Nature, 2024). Is your statement inclined more towards atmospheric chemistry? The discussion may be put in better way and may be supported with references (or this sentence may be removed).

We are certainly aware of the landscape of weather forecasting models built around AI and ML. In fact, there is a pretty big and extensive set of literature here including leading works from DeepMind, Google, Microsoft, Baidu, NVDIA and others.  This gets a bit messy, and a bit beside the point, to start summarizing the different efforts being developed in ML/AI in this climate/weather space. If the reviewers feel strongly about this, we can include comments on these different architectures.  But it should be noted that the results presented take seconds on a simple laptop, not months to train on hundreds or thousands of GPUs.  DMD is quite different in what it as an algorithm.

Thus this statement is specifically aimed at atmospheric chemistry and we are specifically interested in a technique that is a straight regression and does not require the training and computational effort that is currently being deployed by the above companies.

"Both of these limitations make their use in global atmospheric chemistry modeling problematic. Certainly the landscape of models is growing rapidly, with machine learning techniques especially proving useful in weather and temperature forecasting.  These methods are driven by leading tech companies which at scale are training such models with many GPUs over long periods of time to achieve their exceptional performance.  However, a computationally efficient and adaptive ROM approach is embodied by DMD, which is a simple regression requiring no training, cross-validation and hyper-parameter tuning.  It is a straight regression much like a line fit"

l.73-74: "Understanding the composition of atmosphere……(Jacob, 1999)." This is a general introductory text and may be moved somewhere in introduction or may be deleted since similar information and citation has appeared already.

This sentence has been deleted.

l.105: Model output was for 1 year. Why later on about 40/60 days' data have been used for training and validation, to save computation time? How does length of data impact the performance?

As noted in the text, there are some important issues in picking the data for fitting, including that the number of days should have a consistent daylight for the model.  Thus we are not training across multiple years, but rather using the DMD with a sliding window with a relatively stationary distribution in the data.  Thus using summer data to predict winter is not what DMD is good at. But DMD is so cheap, one can simply build a winter model on the fly in seconds. Thus, we have constrained the analysis to one season - we picked 2 months of summer (July - August). Some time in September the day lengths start changing. DMD is sensitive to change in the "period" of data since it is fitting Fourier modes in time, in our case this means the length of a day. Otherwise, the length of data will only increase the computation time accordingly.

l.112: nfeatures = 143 + 91 + 3 + 143 = 380; mention what each number is; (143 is given to be chemical species, what about other numbers here)

Thank you, this has been fixed.  There are other outputs to the GEOS-CHEM code, but they are not used here.

l.117: define abbreviation SVD at first usage

Thank you, we have fixed this.

l.133: "turn-off of dynamics during night times". Do you mean "variability due to photo-chemistry is absent in night". If so, you may re-phrase that way so not to confuse reader with atmospheric dynamics (that is active in night also!).

At the first mention, we do now state that the sunlight is the driver of most of the chemistry in the atmosphere, and thus explain what turn-on and turn-off mean in this context.

l.157-158: Yes, optDMD is clearly better but is there a more quantitative evaluation of the "time evolution" shown in the figure? Is the performance constant with time? How long in time (beyond shown), performance may sustain?

With consistent day lengths, optDMD performance is the same for all time.  Specifically, as noted in comments to the other reviewer, the dynamics can be constructed to be stable for all

time. The forecasting results show how well the performance sustains for the optDMD models. However, as also noted just above, the true failure of the model is that summer and winter have very different daylight hours, thus compromising the DMD model if a summer model is used in winter and vice versa. But again, DMD is simply updated on the fly in seconds and there is no need to try and train models working in all regimes, one simply refits as data is incoming in a computationally efficient manner.

Page 10: Few question marks are appearing after Algorithm. Check and refer correctly. Several places citations are also having some issues which may be proof-read/ corrected.

Thank you, this is fixed.

l.210: Correct "Nitrous Oxide" to "Nitric Oxide" and "Nitrous dioxide" to "Nitrogen dioxide"

Thank you, this is fixed.

l.220: "The results are consistent for all chemical species". What exactly this statement means?

The results presented are for OH, but the same results, let's call it an eye-ball metric, hold for all the chemical species studied. Constrained optDMD model gives the most accurate reconstruction and faithful forecasting. So although we could actually compute an RMS score between model and prediction, this RMS is basically the same for forecasting and reconstruction of all the chemicals.

Is Fig 5 being refereed to?

Both Figures 5 and 6 present the results discussed and referenced in the paper.

Describe rationale behind choosing ranks (r=25 for CONC and r=50 for TEND)?

The truncation is based upon truncating using \textit{hard-thresholding} at a rank $r$ at which the relative error in reconstruction has an elbow, i.e. the error graph flattens out without further decrease.

A scope of improvement is that several figures on results (Fig 7, 8, 9) have limited discussion and that too is qualitative. Explain what authors observe in relative errors (Fig 7) and how they decide on number of modes.

The number of modes is chosen based upon an "elbow" in the singular value spectrum of the SVD of the data. This is a fairly standard way to do rank truncation. A difficulty in discussion the modes is that there are normally 25-30 models linearly combined to construct the solutions.

And unfortunately, our lead domain expert (Keller) left NASA upon completion and submission of the work.  He certainly had the capability of going into great detail on the modal structures and how they represent well-known spatio-temporal trends in the community.  Regardless, the paper is much more focused on the algorithm's ability to produce the these patterns and forecast the future states for different chemicals at different lat, long, elev positions.  As such, the algorithm accomplishes the primary task of the paper.  We have added the following at the beginning of Sec. 2

"Many of the dominant spatio-temporal features of atmospheric chemistry are well-understood through extensive simulation and data collection~\citep{jacob1999introduction,brasseur2017modeling}.  This will not be the focus of this work, but rather a robust, computationally efficient and accurate reduced order model for reconstructing and forecasting the dynamics."

Fig 8, 9: are you comparing these maps with GEOS-Chem based maps for evaluation? Add some quantitative discussion in line with major results.

See above comments

l.276-278: missing spikes are attributed to selection of fewest modes possible. What is the rationale behind not taking few more modes and trying to improve model performance. Air quality spikes are often of significant interests.

Spikes are very difficult for DMD to capture.  This has been observed in various data sets without any satisfactory explanation.  In general, the DMD model "under" predicts the spike-type variations.  We would have to consider a lot more modes to capture the transient spikes (within one time step of 20 minutes) in the data, and we risk picking up more noise that is not representative of the actual chemistry. The errors quantified for forecasting did not indicate to me that we needed to improve the model further. But if this is of interest to study further, and is not just a glitch in the underlying chemistry model, this can be explored. These, at least, are worth exploring further with DMD.  But none of the variants of DMD can seem to get this right.

Fig. 10: the legend STARTData does not seem to be defined in manuscript. Please check.

START needs to be replaced by CONC.  We will fix this figure in order to get this right in the final version.

If possible do include results on ozone (O3) also, similar to that done for CO, NO (Fig 8-9). Ozone is affected by photochemistry as well as transport (and is precursor of OH). Getting O3 distributions right could be very useful.

We are working on getting those figures together in order to show the results.  This will be done shortly and uploaded to the final version.

References: First reference "Global modeling of tropospheric……." Is incomplete, please add names of authors.

Thank you, this has been fixed.

References

Arcomano, T., Szunyogh, I., Pathak, J., Wikner, A., Hunt, B. R., & Ott, E. (2020). A machine learning-based global atmospheric forecast model. Geophysical Research Letters, 47, e2020GL087776. https://doi.org/10.1029/2020GL087776

Kochkov, D., Yuval, J., Langmore, I. et al. Neural general circulation models for weather and climate. Nature 632, 1060–1066 (2024). https://doi.org/10.1038/s41586-024-07744-y